# Light-gated redox switching and actuation in polymer hydrogels

Roza R. Weber[1], Robert Hein [1,2] ✉, Alexander Ryabchun [1], Yohan Gisbert[1,4], David Garcia Romero[3], Maria Antonietta Loi [3] & Ben L. Feringa [1,3] ✉

Stimuli-responsive materials based on molecular switches, introducing life-like properties such as adaptive behavior in an aqueous environment, are fascinating, providing numerous opportunities to control functions and enable future applications like actuators and soft robotics. Light-responsive molecular systems are receiving particular attention, due to the non-invasive stimulus and distinctive spatio-temporal control possible with photoswitches. In contrast, redox-switching is quantitative, non-volatile and associated with significant changes in material properties, but lacks spatio-temporal precision. Herein we address this challenge in the first proof-of-principle demonstration of light-gated redox switching of polymer hydrogel materials, thereby combining the advantages of both strategies. We present a unique approach where irradiation controls the intrinsic redox properties of the system. This is enabled by the reversible and versatile light- and redox-responsive bisthioxanthylidene switch embedded in a polymer hydrogel, whose two-electron oxidation potential is strongly modulated by light. As a result, oxidation of the material, which is associated with large changes in color, fluorescence, swelling and actuation can be carried out in water with high precision in space and time by photo-masking. This light-gated redox-patterning of the material can be exploited for numerous functions including, as demonstrated here, complex motion and reversible surface texturing.

Taking inspiration from living systems that can adapt and respond to external signals, stimuli-responsive materials have emerged as indispensable tools in numerous applications. Their ability to act upon various external stimuli by changing, among others, their structure, optical/electronic properties or by inducing motion offers fascinating opportunities in wide-ranging scenarios, including drug delivery, tissue engineering, sensors, responsive surfaces and actuators[1–5]. Besides the challenge to operate in water, the ability to undergo reversible shape changes is paving the way for adaptive, biomimetic systems and soft robotics[6,7].

In this context, stimuli-responsive polymeric materials, including networks and (hydro)gels, have received particular attention, as their chemical constitution, morphology and mechanical properties can be easily adjusted by variation of the (co)monomer building blocks, degree of crosslinking, or polymerization conditions[8–12]. Many hydrogels are not only highly tunable and biocompatible but also (inherently) responsive to external stimuli such as pH or temperature, which can lead to reversible phase and gel-sol transitions or swelling/deswelling and associated functions[13–17]. Another attractive approach to render various materials stimuli-responsive is the use of photochemical molecular switches and motors as versatile and chemically tunable responsive elements[18–22]. Their integration into materials, including polymeric networks and hydrogels, has enabled the efficient translation of nanoscale molecular switching functions to the

[1]Strathingh Institute for Chemistry, University of Groningen, Groningen, The Netherlands. [2]Organic Chemistry Institute, University of Münster, Münster, Germany. [3]Zernike Institute for Advanced Materials, University of Groningen, Groningen, The Netherlands. [4]Present address: Univ Rennes, CNRS, ISCR – UMR 6226, F-35000 Rennes, France. ✉e-mail: robert.hein@uni-muenster.de; b.l.feringa@rug.nl

macroscale[19,21,23–35]. Due to its high spatio-temporal control and non-invasive, waste-free nature, light is the most commonly used stimulus to control such switchable materials, enabling a wide range of complex functions, including directional motion[36,37]. Nevertheless, photo-switching is typically volatile, not quantitative and often associated with comparably small changes in molecular properties (e.g., no change in charge)[38,39]. A notable exception to this paradigm is the spiropyran class of photoswitches[40], which undergoes significant changes in polarity/charge upon light-induced switching, which has been exploited in various stimuli-responsive materials. However, when the material is no longer irradiated, this photoswitch typically rapidly thermally relaxes back to its ground state, thereby limiting applications in which a longer-lasting response to a stimulus is sought[24,41]. Overall, the effects of photoswitching on material properties are usually limited in magnitude and/or temporal stability/longevity. The incorporation of redox-active building blocks into materials can in principle, address these shortcomings, as redox-switching is generally quantitative and associated with large-scale, non-volatile changes of charge and associated properties[42–48]. However, controlling such redox switches with high spatial precision is challenging, and their chemical stability in different charge/oxidation states is often compromised, especially in aqueous environments and in the presence of air. The highly attractive combination of both photo- and redox-switching in a single material could merge the advantages of both but remains virtually unexplored. Photoredox switching and patterning of gels has recently been demonstrated, utilizing an extrinsic photo-redox catalyst[47–49], but modulating intrinsic redox properties non-invasively by light in aqueous media would offer unique opportunities. Herein, we address this challenge in reporting a proof-of-principle example of integrated light-gated redox switching in a polymeric hydrogel network, not by using photo-redox, but through the use of a molecular switch. Using the principle of light-gated redox switching, we take advantage of the intrinsic high spatiotemporal control of light-driven switching, as well as the large amplitude, quantitative and non-volatile nature of redox-driven switching. This was enabled by the highly versatile, multi-state and multi-stimuli-responsive overcrowded alkene bisthioxanthylidene (BTX, Fig. 1a). In its neutral ground state, this switch is colorless, blue-fluorescent and adopts an *anti*-folded conformation, which upon UV light irradiation quantitatively converts to a metastable *syn*-folded, colorless, non-fluorescent state[50–53]. Two-electron oxidation of either of the neutral switch states affords the orthogonal, dicationic state, which is strongly purple-colored and red-fluorescent[50]. Importantly, this oxidation is not only associated with these significant changes in charge, optical properties and geometry but also proceeds without the build-up of radical cation intermediates and is associated with a high degree of reversibility[50,54]. Another distinctive and crucial property of the BTX switch is the significant dependence of its oxidation potential on the geometry of the neutral state; the *syn*-folded state is significantly easier to oxidize than the *anti*-folded state. We exploit this unique feature here to enable highly localized and patterned oxidation by irradiation of a BTX-containing hydrogel in the presence of a weak oxidant; only where the gel is photo-switched to the *syn*-folded state does oxidation to the dication occur. This process is not only highly reversible under ambient and aqueous conditions but is also associated with significant (patterned) optical and charge changes and associated swelling and actuation (Fig. 1b, c).

## Results and discussion
### Hydrogel preparation
In order to enable BTX integration into hydrogels (Fig. 1C), a polymerizable BTX monomer (**BTX-MA**), containing a methacrylate motif, was synthesized in three steps from 2,2′-dimethoxy-BTX[51] as shown in Fig. 1d. From this, a hydrogel (**BTX-gel**) was prepared by co-polymerization of **BTX-MA,** *N,N*-dimethylacrylamide (DMAAm) as

comonomer, *N,N*′-methylenebisacrylamide (MBAm) as crosslinker and IRG819 as photoinitiator in anisole. Unless otherwise noted, a ratio of 10:89:0.7, **BTX-MA**/DMAAm/MBAm was used, which was optimized based on the oxidation-induced swelling and stability of the material (see "Methods" section and Supplementary Fig. 34 for further details).

In this manner, hydrogel samples of three different types/morphologies were prepared as illustrated in Fig. 2. Specifically, irradiation of the monomer blend with 455 nm light, which is not absorbed by any of the monomers, induces homogeneous polymerization resulting in flat, free-standing hydrogels (Method 1). In the same manner, homogeneous, but solid-supported gels were prepared by polymerization on top of 3-(trimethoxysilyl)propyl methacrylate (3-MPS)-functionalized glass or ITO, enabling covalent immobilization onto these substrates and the formation of much thinner gels (Method 3). Alternatively, in Method 2, free-standing polymer gels were prepared by polymerization induced by 365 nm light, which is strongly absorbed by **BTX-MA**, leading to a significant attenuation of the light dose on the side of the mold that is facing away from the LED. As a result, the degree of polymerization and crosslinking is significantly higher on the irradiated side and follows a gradient. Regardless of the specific method, after photo-induced polymerization, the gels were removed from their glass mold, cut into the desired shape/size, and the solvent was exchanged first to ethanol and then to water. In the case of the isotropic films prepared by Method 1, this solvent exchange from anisole to water induced gel swelling by 49%. Gelation was further confirmed by rheology, revealing a storage ($G'$) and loss modulus ($G''$) of $3 \pm 0.1$ kPa and $0.4 \pm 0.02$ kPa, respectively. For the hydrogels containing a density gradient (Method 2), different 3D shapes were obtained depending on their dimensions[55]. For example, for samples with a large length-to-width ratio, curling into helices was observed, with no preference for left- or right-handedness (Fig. 2B). In contrast, when the length-to-width ratio was closer to one, short-side rolling was mainly observed.

### Light switching of the hydrogel
To investigate the influence of BTX integration into the hydrogel matrix on its switching properties, optical switching studies were first carried out. To this end, thin, glass-supported **BTX-gel** of sufficiently low optical density were prepared by Method 3 and studied by UV/Vis spectroscopy in water. As shown in Fig. 3B, irradiation of the gel with 365 nm light induced rapid and notable changes in the gel's absorbance, with distinct isosbestic points indicating selective interconversion between the two photoisomers. Specifically, the initial lowest-energy band at $\lambda_{max} = 370$ nm, which can be ascribed to the BTX in the *anti*-folded state, entirely vanishes in favor of a new, higher energy band at $\lambda_{max} = 322$ nm, which is in excellent agreement with light-driven switching to the *syn*-folded state[50]. This metastable state quickly thermally reverts to the initial *anti*-folded state with a half-life time of $t_{1/2} = 150$ s at 25 °C, which is very similar to that of the **BTX-MA** monomer in CH$_3$CN solution ($t_{1/2} = 158$ s), highlighting that interconversion between the *syn* and *anti*-folded states proceeds without impediment in the aqueous gel environment (Supplementary Fig. 9). This switching process is not only highly reversible over at least 11 cycles (Supplementary Fig. 8), but is also associated with a notable change in the fluorescence of the gel; in the *anti*-folded ground state, the material is strongly blue-fluorescent while photoswitching to the *syn*-folded state induces virtually complete fluorescence quenching (Supplementary Fig. 51). The bright blue fluorescence before irradiation, which becomes very weak under continuous irradiation, can be observed with the naked eye. Unless explicitly stated otherwise, all further discussions refer to the BTX switch in its native, *anti*-folded state.

### Redox switching and actuation of the hydrogel
The electrochemical properties of the **BTX-MA** monomer expectedly mirror those of similar BTX derivatives when dissolved in organic

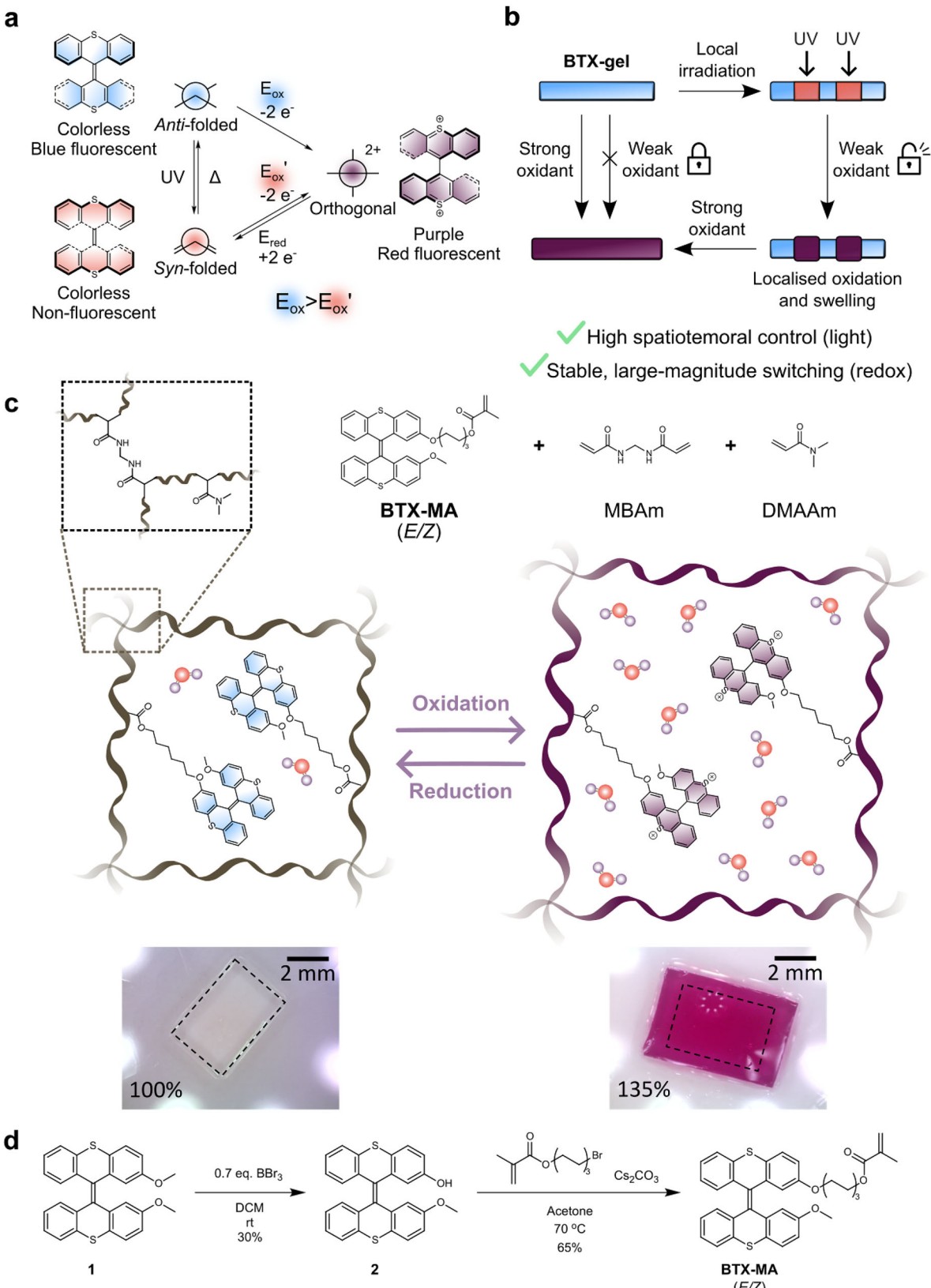

**Fig. 1 | Overview of BTX switching and its integration into hydrogels. a** Multi-state redox- and light-driven switching of BTX. **b** Schematic representation of the combination of redox- and light-driven switching of a BTX-containing hydrogel enabling a unique exploitation of the advantages of both approaches. Specifically, light-gated redox switching is associated with high spatio-temporal control *and* non-volatile large-magnitude actuation. **c** Schematic representation of **BTX-gel**, its components, and its optical and swelling response upon redox stimulation. **d** Synthetic scheme of the synthesis of **BTX-MA**.

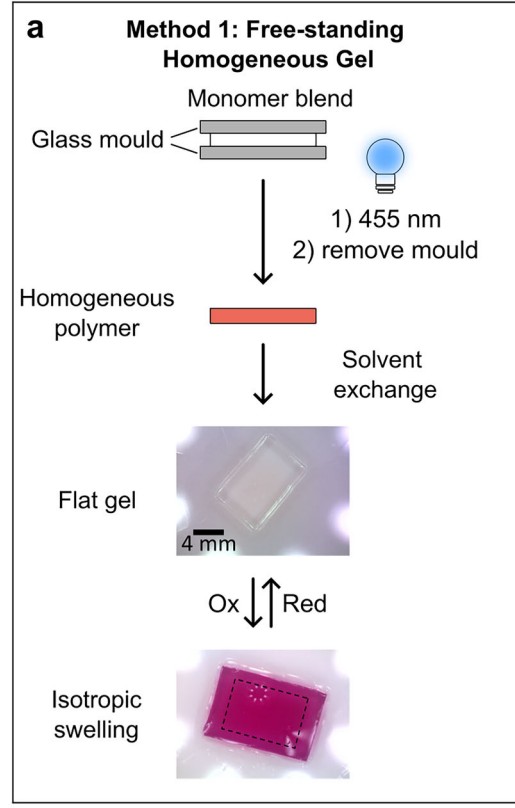

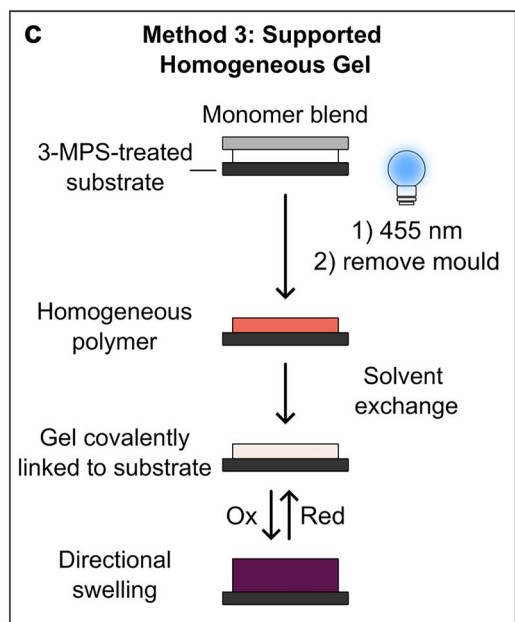

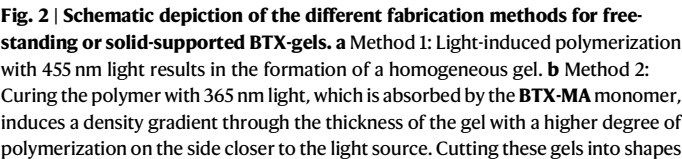

**Fig. 2 | Schematic depiction of the different fabrication methods for free-standing or solid-supported BTX-gels. a** Method 1: Light-induced polymerization with 455 nm light results in the formation of a homogeneous gel. **b** Method 2: Curing the polymer with 365 nm light, which is absorbed by the **BTX-MA** monomer, induces a density gradient through the thickness of the gel with a higher degree of polymerization on the side closer to the light source. Cutting these gels into shapes with different aspect ratios gives rise to various 3D shapes. **c** Method 3: Polymerization with 455 nm light, but using a substrate functionalized with 3-MPS enables the covalent attachment of the gel to the substrate. Depending on the method used, chemical oxidation of the gels with CAN results either in isotropic swelling (**a**), uncurling or unwinding of 3D shapes (**b**) or swelling mostly in the direction perpendicular to the substrate (**c**).

solvent ($CH_2Cl_2$), as resolved by cyclic voltammetry (CV). This most notably encompasses simultaneous two-electron oxidation at high potentials (+0.80 V vs. Fc/Fc$^+$ at $v = 100$ mV s$^{-1}$), corresponding to conversion from the neutral *anti*-folded to the orthogonal dicationic state (Fig. 3d)[50,51]. Transfer of two electrons upon oxidation was experimentally supported by Shoup-Shabo analysis[56] of the

chronoamperometric response of a micro-disk electrode ($r = 5 \mu$m), as detailed in the SI, Section S4.2, Supplementary Figs. 28 and 29. Additional evidence for simultaneous two-electron oxidation was also obtained by bulk controlled-potential electrolysis, which confirmed that both the oxidation and the reduction waves comprise two electrons (Supplementary Fig. 30). As discussed in more detail in the SI

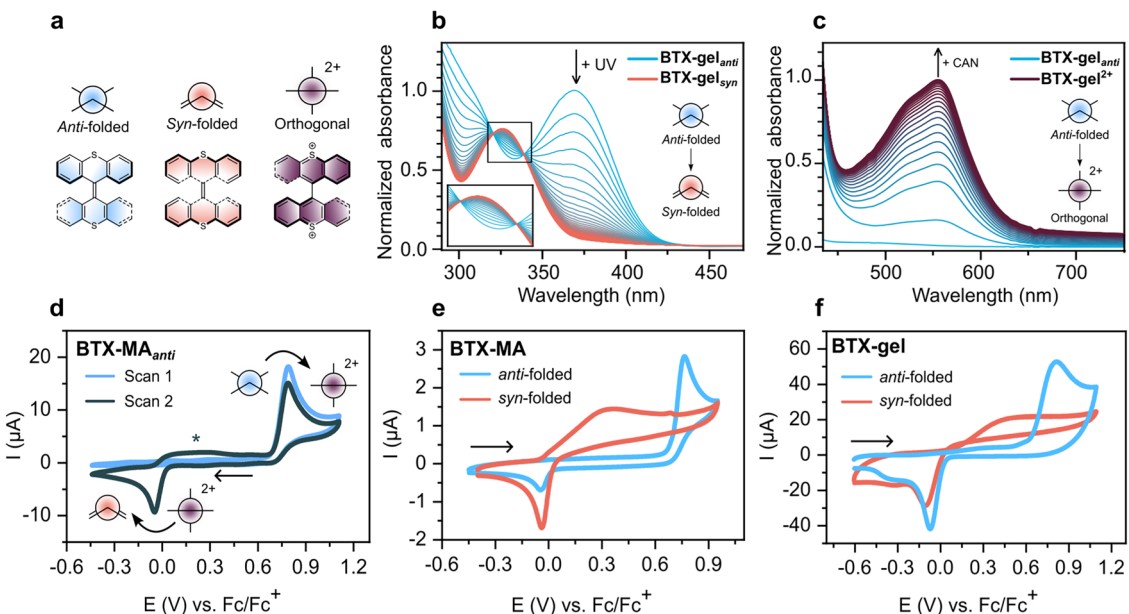

**Fig. 3 | Light- and redox-induced switching of BTX. a** Schematic depiction of all three switching states of BTX. **b** Change of absorbance of **BTX-gel** upon irradiation with 365 nm light in water at 25 °C. **c** Change of absorbance of **BTX-gel** upon oxidation to **BTX-gel²⁺** by chemical oxidation with 15 mM CAN in water. **d–f** CVs of **BTX-MA** and **BTX-gel** (on ITO) in CH₂Cl₂, 100 mM TBAPF₆. The black arrows indicate the starting point and direction of the first scan. For solution-phase studies, a glassy carbon (GC) working electrode was used (see also Supplementary Fig. 13 for CVs measured using Pt). **d** 0.5 mM **BTX-MA** at *v* = 100 mV s⁻¹. The absence of a

reduction peak in the first (light blue) scan illustrates the hysteretic, dynamic nature of the redox switching; unless the dicationic switching state is first (electro-chemically) generated, no reduction is observed. The peak marked with an asterisk arises from the oxidation of the *syn*-folded state. **e** 0.25 mM **BTX-MA** at *v* = 25 mV s⁻¹. **f BTX-gel** on ITO at *v* = 100 mV s⁻¹. The *syn*-folded state, obtained by continuous, in situ irradiation of the solution with 365 nm light, displays a significantly more cathodic oxidation potential (**e** and **f**).

Section S4.2, and in the literature[57,58], the two-electron oxidation from the *anti*-folded state to the orthogonal dicationic state proceeds via an $E_1CE_2$ mechanism with strong potential inversion, i.e., $E_2 < E_1$ (Supplementary Fig. 15), with a very fast chemical rearrangement in the intermediate radical cation state. This was supported by calculations of the optimized geometries of all relevant isomers and their relative energies (Supplementary Figs. 16–27). These theoretical investigations also confirmed that both the *E* and *Z* isomers of the BTX-switch display very similar geometries and energies and are thus expected to switch in the same manner.

As a result of the significant geometric rearrangements that are associated with electron transfer, the redox process displays a very high degree of hysteresis; two-electron reduction of the dication back to the neutral state occurs at much more cathodic potentials (−0.05 V *vs.* Fc/Fc⁺ via an EEC pathway, which first quantitively forms the *syn*-folded state, Supplementary Fig. 15), in good agreement with other BTX analogs[50,51]. This behavior is also largely retained upon integration into the hydrogel. Specifically, **BTX-gel** immobilized on an ITO electrode (via Method 3) and swelled in CH₂Cl₂ displays redox properties that are qualitatively analogous to the monomer, with a very similar oxidation peak potential of + 0.82 V *vs.* Fc/Fc⁺ (Fig. 3f).

Even in aqueous electrolyte the voltammetric signatures of **BTX-gel** are not significantly altered (Supplementary Fig. 14), revealing that BTX is a versatile and reversible photo- and redox-responsive switch not only in organic solvents but also in aqueous media.

This was further confirmed by the chemical oxidation of **BTX-gel** in water with the strong oxidant ceric ammonium nitrate (CAN), which induced a color change from transparent to dark purple (Fig. 1c). This is also reflected in the appearance of a new absorption band at 555 nm (Fig. 3c), which is in excellent agreement with the generation of the dicationic BTX oxidation state[50]. Importantly, by reduction of **BTX-gel²⁺** with sodium ascorbate, the BTX²⁺ absorbance band fully vanishes, and the initial UV/Vis spectrum of the neutral **BTX-gel** is recovered

(Supplementary Fig. 10A, B). This redox switching sequence can be carried out over multiple cycles with a relatively high degree of reversibility for redox switching under ambient, aqueous conditions (see SI Supplementary Fig. 10C)[45,46]. The relatively high fatigue-resistance, compared to some other redox-responsive materials[45,46] can be attributed to the rare two-electron redox process of the BTX switch, which enables interconversion between two diamagnetic closed-shell and chemically stable switching states (BTX and BTX²⁺), without build-up of intermediate, reactive radical cation states. In analogy to the photo-switching process, this redox stimulation of the gel is also associated with significant changes in fluorescence; upon oxidation, the initial strong blue fluorescence disappears in favor of a weaker, orange-red fluorescence (Supplementary Fig. 52), which also confirms near-quantitative conversion to the BTX²⁺ oxidation state. However, in contrast to photoswitching, which induces no notable morphological change in the gel, redox switching does induce significant swelling of the gel (Fig. 2a–c). This is consistent with electrostatic repulsion between the switches as well as the larger hydrophilicity of the oxidized, highly charged gel state, which results in significant uptake of more water into the gel (Donnan effect), both of which are potentially also tunable by changing solvents/ion concentrations (*vide infra*). As a result, a significant increase in the characteristic length (sum of the diagonals) of the homogeneous **BTX-gel** of 35 ± 3% (corresponding to an increase in volume of 146%), is observed and the gel softens, which is also reflected in a decrease of the storage modulus (*G'*) by 24% (Supplementary Fig. 33). To put this value into context, previous work on redox-active hydrogels, such as the materials developed by Harada and coworkers in 2013, which are based on reversible host-guest interactions to swell and shrink, showed an oxidation-induced swelling of only 11% in length[46]. Other representative redox-responsive gel actuators were reported in 2023 by Ikeda, which were based on fol-damers. These organogels, which were not operated in water but in acetonitrile, swelled up to 25% when oxidized[45]. A different approach

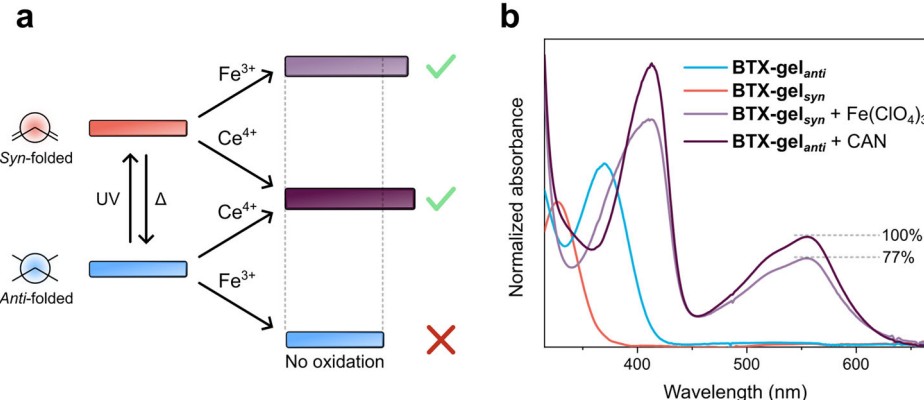

**Fig. 4 | Light-gated redox switching of BTX-gel. a** Schematic depiction of the principle of light-gated oxidation of **BTX-gel** in water. The strong oxidant CAN can oxidize the BTX-gel in both *anti-* and *syn*-folded states. In contrast, the weaker oxidant Fe(ClO₄)₃ is only capable of oxidizing the BTX motifs in the *syn*-folded state, *i.e.*, when the gel is irradiated, thereby enabling localized light-gated redox-switching. **b** UV/Vis spectra of the **BTX-gel** in water under different conditions, confirming the successful oxidation to BTX²⁺ in the presence of Fe(ClO₄)₃ *and* light.

to generate responsive hydrogels, which is also based on light and redox stimuli, makes use of photoredox catalysts. This strategy was notably developed by Barnes and coworkers, wherein irradiation of a separate photocatalyst induced local reduction and shrinking of viologen-foldamer gels[47,49,59]. However, this had to be performed in an oxygen-free environment and induced a slightly smaller decrease in volume of 88%, which corresponds to a 23% decrease in length[49]. Exposure to oxygen then induced oxidation and led to the recovery of the initial gel size. Of additional note is that in all of the examples discussed above, only 2-3 full redox switching cycles were shown.

This large degree of redox-induced swelling observed herein can be exploited in the construction of various actuators. For example, oxidation of free-standing **BTX-gels** containing a density gradient (Method 2, Fig. 2b) induces more swelling on the side of the material that has a higher BTX density, which in turn induces a motion that counteracts the initial bending moment. Depending on the specific shape of the material, this results in different motions, including, for example, flattening of an initially curled tube or extension of a helical spring (Fig. 2b, Supplementary Figs. 25, 26, and Supplementary Movies 1 and 2).

Importantly, this swelling and the resulting actuation is highly reversible; upon reduction with aqueous ascorbic acid shrinking of the gel back to its original size and shape was observed. Furthermore, successful oxidation was achieved in a wide variety of aqueous environments containing different counter anions/salts (NaPF₆, NaNO₃ or NaClO₄) at different concentrations (0–100 mM) and different *pH* (1–7), see Supplementary Figs. 41–43 and associated discussions. In general, we observed no strong influence of differing conditions on the swelling magnitude, which was around 20–30% in all cases. The only consistent, but weak trend we identified is that a higher ionic strength induces somewhat diminished responses, which is in good agreement with Donnan theory[60]. These observations suggest that swelling is primarily driven by the generation of the dicationic charge state, that specific ion pairing effects (i.e., the nature of the counter anion) are not overly important and that the BTX gel actuators can be used under various aqueous conditions and in different application scenarios. Moreover, as shown in Supplementary Fig. 34, adjusting the amount of **BTX-MA** or crosslinker has a comparably much larger effect on the observed magnitude of redox-induced swelling, but can also alter the inherent mechanical properties of the gel.

Lastly, this redox-switching does not only lead to large-amplitude change of the material properties (color, stiffness, shape) but is, in contrast to photoswitching, non-volatile. As a result, the oxidized **BTX-gel²⁺** state can, in principle, be maintained indefinitely; by the naked

eye we observed only minor changes in color/morphology of the **BTX-gel²⁺** after one month in aqueous solution (Supplementary Fig. 44).

## Light-gated Redox Switching, Patterning and Actuation

A feature of the BTX switch is, as discussed above, the significant dependence of its redox properties on its conformational state; upon light-driven switching of the *anti*-folded to the *syn*-folded state, a significant cathodic voltammetric shift of the oxidation potential is observed[50]. Indeed, for the **BTX-MA** monomer in CH₂Cl₂, an at least 140 mV lower oxidation potential was observed by UV light-induced switching to the *syn*-folded state (Fig. 3E and Supplementary Fig. 13). As expected, this behavior was also observed for **BTX-gel** (Fig. 3F). The exact origin of the change of oxidation potential between the *anti-* and *syn*-folded states has not yet been elucidated. We initially hypothesized that the facilitated oxidation for the *syn*-folded state is related to an increase in its HOMO energy and thus ionization potential. However, DFT calculations at the ωB97X-D4[61,62]/def2-TZVPD[63] level of theory suggest that the difference between HOMO energies is too small to explain the observed magnitude of potential shift (see Supplementary Figs. 31, 32 and Supplementary Tables 2, 3 and associated discussions). This indicates that other mechanisms, e.g., related to the kinetics of conformational rearrangements, are relevant.

Nevertheless, this rare property enables the local light-gated oxidation of the **BTX-gel**. Specifically, in water, the weaker oxidant Fe(ClO₄)₃ is incapable of oxidizing the BTX switch in the *anti*-folded state, but is sufficiently strong to generate BTX²⁺ from the neutral *syn*-folded state (Fig. 4a). As such, a **BTX-gel** soaked in an aqueous solution of Fe(ClO₄)₃ will not oxidize unless irradiated. Indeed, no swelling or color change were observed even on prolonged exposure of the gel to this weaker oxidant. However, irradiation of the gel with 365 or 395 nm light in the presence of Fe(ClO₄)₃ induces rapid oxidation as evidenced by the characteristic purple coloration and swelling.

Importantly, this photoswitching-oxidation cascade results in an almost identical final switching state, that is the oxidized **BTX-gel²⁺**, as evidenced by UV/Vis spectroscopy (Fig. 4b). Specifically, the absorbance spectrum of the **BTX-gel²⁺** obtained by light-gated redox is qualitatively identical to that obtained by purely chemical redox with CAN, albeit with ~20% smaller absorbances, indicative of a slightly lower degree of conversion to the dicationic switching state. This is most likely the result of the somewhat limited degree of UV light penetration throughout the whole sample, such that parts that are further inside the gel are not as efficiently converted to the dication. As a result, the degree of swelling of such gel samples is also slightly lower (23 ± 3%) upon light-gated oxidation. However, this effect is expected

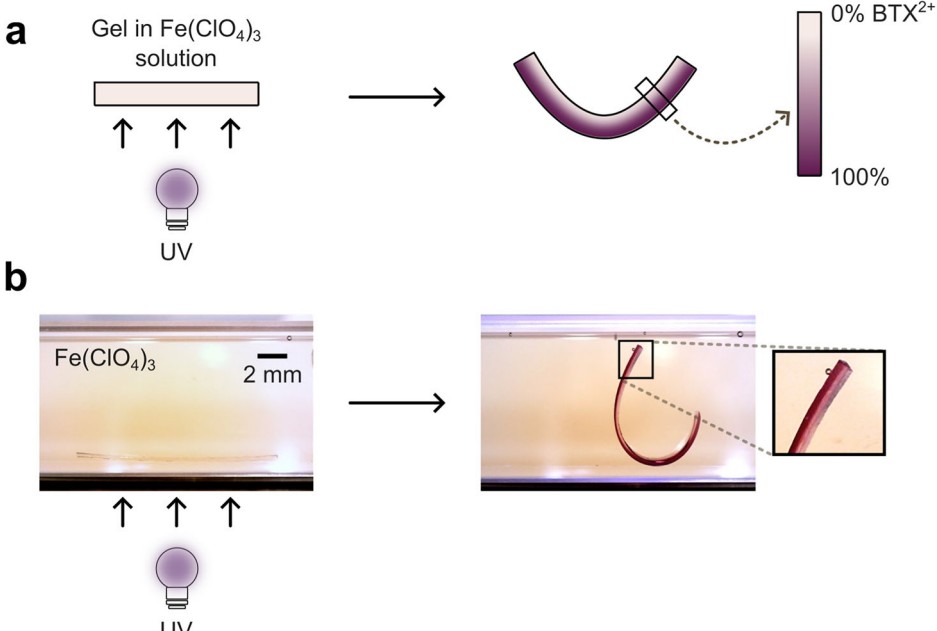

**Fig. 5 | Curling of BTX-gel strip by light-gated redox switching. a** Schematic illustration and (**b**) photographs of light-gated-redox actuation of an initial homogeneous **BTX-gel** (Method 1) in an aqueous solution containing 30 mM aqueous $Fe(ClO_4)_3$ by continuous irradiation with 365 nm light from the bottom until the bending motion stopped. This induces a gradient with a higher degree of oxidation, and thus swelling, on the side of the material that is closer to the light source, which ultimately results in significant bending.

to be strongly dependent on, for example, the shape/thickness of the material and light dose, such that very thin or less dense gels might be converted to the dicationic state to an even higher degree. Therefore, this inner-filter-induced attenuation of light penetration can again be taken advantage of in the generation of oxidation gradients and thus highly localized swelling and actuation. For example, irradiation of an initially homogeneous **BTX-gel** in an aqueous solution of $Fe(ClO_4)_3$ from one side induces an oxidation and thus swelling gradient and results in the bending of the sample away from the light (Fig. 5, Supplementary Fig. 26 and Supplementary Movie 3). The samples reach their maximum curvature after approximately 10 min (Supplementary Figs. 48 and 49). Importantly, such a gradient can be reversibly generated in an otherwise homogeneous sample and hence requires no specific preparation of gels with an inherent crosslinking imbalance. Furthermore, the extent of light-gated redox switching and the resulting swelling/motion can be controlled by adjusting the irradiation duration. For example, as shown in Supplementary Fig. 50, three different levels of bending were achieved by interrupted irradiation with short bursts of light. Notably, there is a lag period between the short irradiation and the full extent of the resulting actuation, which can be attributed to the comparably slower diffusion-controlled redox/swelling process. Nevertheless, these results indicate that any desired degree of light-gated redox switching can be achieved by controlling the overall photon flux.

More generally, these findings highlight the power of this intrinsic light-gated redox switching as it enables the combination of the unique advantages of both light and redox modulation approaches. Specifically, redox switching is associated with large-scale, non-volatile modulation of various properties, including charge, color and fluorescence and related material properties (softness/swelling), but is very difficult to carry out with high spatiotemporal control, which is one of the major advantages of photoswitching as shown here.

The high-spatiotemporal control of the oxidation process also enables facile and reversible redox patterning of the hydrogel by photo-masking, as shown in Supplementary Fig. 54. In this manner, a wide variety of precise redox patterns can be created, which is, as discussed above, also associated with (local) control over color, fluorescence, and swelling response. Free-standing hydrogels can be patterned with detailed images, for example, a windmill, as shown in Fig. 6a. In this manner, the irradiated area becomes strongly purple-colored, with weak orange-red fluorescence. In contrast, the non-irradiated areas remain colorless with bright blue fluorescence, thereby enabling visualization of the pattern both under ambient light as well as via fluorescence. As a result of the local swelling response, a notable corrugation of the material is also induced (Fig. 6a). Using the solid-supported gels (Method 3), this phenomenon can be further exploited as the swelling induced by patterned oxidation is restricted to expansion in the z-dimension, thereby inducing well-defined surface textures (Fig. 6b). For example, light-gated redox patterning with a mask enables the generation of various surface patterns such as our university logo or striped undulations, as elucidated by 3D-profilometry (Fig. 6c and Supplementary Fig. 53). This also shows that this patterning not only generates notable textures with peak-to-valley distances of up to 120 μm, but is also reversible, enabling facile repatterning of polymer hydrogels with a different design. As such, this approach may also enable the transport of macroscopic cargo across a surface.

Light-gated redox switching was demonstrated as a powerful approach for the actuation of hydrogels, combining the advantages of light-switching (high spatio-temporal control) with redox-switching (large magnitude, non-volatile changes). This was enabled by the unique properties of the dual-functional, versatile and reversible three-state BTX switch, whose oxidation potential is highly sensitive to its conformational state, which in turn can be easily controlled by light irradiation. In the presence of $Fe^{3+}$ as a weak oxidant, the **BTX-gel** in its native, *anti*-folded switch state is not oxidized, while (localized) irradiation induces conversion to the *syn*-folded conformer and rapid oxidation to **BTX-gel$^{2+}$**, which is associated with a large range of macroscopic material changes. This includes not only changes in color and fluorescence but also significant swelling due to the ingress of

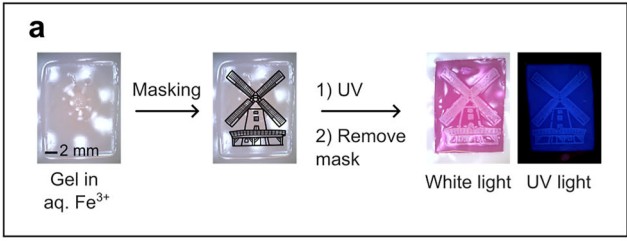
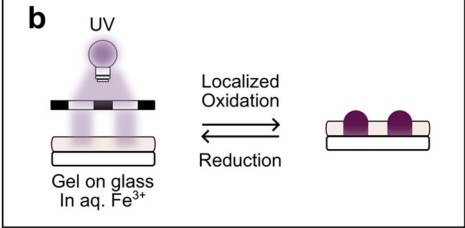

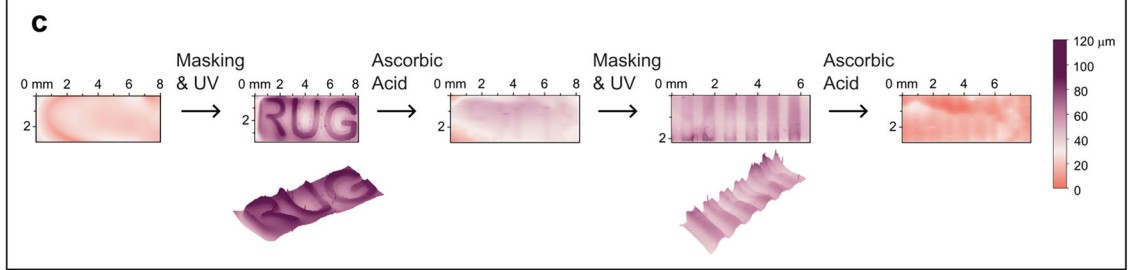

**Fig. 6 | Light-gated redox patterning of BTX-gel in aqueous Fe(ClO₄)₃ with UV light enables high spatio-temporal control over local oxidation of the gel and associated changes in color, fluorescence and swelling. a** Patterning of a windmill onto a piece of freestanding **BTX-gel**. "Windmill" icon by Hey Rabbit from Noun Project CC BY 3.0. **b** Schematic representation and (**c**) 3D profile of reversible surface texture patterning of a substrate-confined **BTX-gel**. Irradiation was carried out for ~30 min to ensure full conversion and the largest extent of swelling. However, in principle, any intermediate swelling state can be accessed by shorter irradiation duration.

water as a result of the generation of the doubly-charged switch state. It should also be noted that this intrinsic light-gated redox switching approach differs conceptually from photo-redox actuation of materials in that all required functions are built-in to a single switching component. As a result, each switching function (light or redox or light-gated redox) can be individually carried out and further tuned by chemical modification of the (BTX) switch[57], for example, such that weaker, less toxic oxidants – ideally ambient oxygen – can be employed. Furthermore, we expect this approach to be extendable to other material classes and switch scaffolds in which light- and redox-switching are co-dependent. Based on the findings presented here, various complex surface textures or macroscopic actuators in aqueous conditions can be readily generated, thereby paving the way for numerous future applications, including soft robotics.

## Methods

### Materials
All reagents and solvents were purchased in high purity from BLDPharm, Sigma Aldrich, TCI Europe, or Acros Organics, and were used without further purification. Dry solvents were obtained from an MBraun SPS-800 solvent purification system. 2-methoxy-9H-thioxanthene-9-thione and 6-bromohexyl methacrylate were synthesized following literature procedures[51,64].

### Hydrogel preparation
A blend of 89 wt% *N,N*-dimethylacrylamide (DMAAm), 10 wt% BTX-MA, 0.7 wt% *N,N*-methylenebisacrylamide, and 0.3 wt% photoinitiator IRG819 were dissolved in anisole to give a 50/50 mixture of solvent and active components (by weight). This solution was injected using a pipette into a glass mold of a desired size with PVC, Teflon or glass bead spacers with thicknesses of 100, 63, or 18 μm, respectively.

"Blank" gels without any BTX were prepared from a blend of 99 wt% DMAAm, 0.7 wt% *N,N*-methylenebisacrylamide, and 0.3 wt% photoinitiator IRG819.

**Method 1.** The monomer blend was injected into a glass mold with 100 μm spacers and cured with a 455 nm LED (Thorlabs M455L4, 54 mW cm⁻²) for ten min to prepare a homogeneous film. After polymerization, the polymer was removed from its mold and cut into the desired shape(s).

**Method 2.** The monomer blend was injected into a glass mold with 100 μm spacers and cured using a 365 nm LED (Thorlabs M365L3, 0.38 mW cm⁻²) for 30 min to create a crosslinking gradient across the thickness of the polymer.

**Method 3.** The glass or ITO slides (glass: 13 × 30 x 1 mm, ITO: 8 × 27 x 1 mm, from ALS Japan) were functionalized with 3-(trimethoxysilyl)propyl methacrylate (3-MPS) following a reported procedure[64].

The functionalized slides were then used as the top half of the molds used for polymerization, and the bottom half was a piece of untreated glass covered with transparent polyvinylchloride (PVC) plastic to prevent the sticking of the gel to the mold. The spacers for these samples were 18 μm (for spectroscopic studies), 63 μm (for the fatigue study) or 100 μm (for profilometry). The polymerization procedure was otherwise identical to Method 1.

After polymerization, the polymer pieces, regardless of the method, were placed in ethanol for at least thirty min, after which the solvent was removed and fresh ethanol was added. This process was repeated 3 x, followed by the same washing process in water.

### Light-switching of the hydrogel: UV/Vis and fluorescence
The width of the glass slides used for all UV/Vis and fluorescence studies with the gels prepared via Method 3 was around 1.3 cm, to fit diagonally in 1 × 1 cm plastic cuvettes. This ensured that the same spot on the gel is measured in every measurement. To study the photo-switching of BTX in the gel, the sample was irradiated in the UV/Vis spectrometer (Agilent 8453 UV/Vis Diode Array System, equipped with a Quantum Northwest Peltier temperature controller) or, for the fluorescence studies, ex situ, using 365 nm light (Thorlabs M365L3, 1000 mA). The fluorescence measurements were performed using the neutral gel, the gel at PSS after irradiation with 365 nm light and the gel after oxidation (15 mM aqueous CAN for 30 min). Fluorescence spectra were recorded on an Edinburgh Instruments FS5 spectrofluorometer. For the excitation spectra, the emission at the maximum emission

wavelength was followed, 450 nm for the neutral gels (both in their native state and at PSS) and 601 nm for the oxidized gel (scan slit 5 nm, detector slit 1 nm). The excitation wavelength for the emission spectra was chosen to be the maximum of the excitation spectrum, 365 nm for the neutral gels (scan slit 5 nm, detector slit 1 nm), and 350 nm (scan slit 2.4 nm, detector slit 1 nm) or 490 nm (scan slit 1 nm, detector slit 3 nm) for the oxidized gel.

## Chemical oxidation and reduction: UV/Vis and fluorescence

Samples (Method 3, on glass) were placed in the cuvette at a 45° angle. First, the absorbance or fluorescence of the neutral BTX-gel was measured. Then, to oxidize the hydrogels, they were submerged in a 15 mM solution of CAN for 30 min, followed by extensive washing with water (submersion in water 3x, rinsing with water) and the spectrum was measured again. To reduce the hydrogels, they were submerged in 15 mM ascorbic acid in water for 15 min, followed by submersion in water (3x) and rinsing with water. For the redox fatigue study, the oxidation with CAN and reduction with ascorbic acid were repeated five times with a gel made using 63 μm spacers. The absorbance spectrum was recorded after each step.

For the light-gated oxidation, 30 mM iron(III)perchlorate in water was added to the cuvette containing the gel on glass. The samples were irradiated in the UV/Vis spectrometer using 365 nm light (Thorlabs M365L3, 1000 mA), or it was irradiated ex-situ at 5 °C for one hour while recording the absorbance spectrum every five min to follow the change in absorbance.

## Chemical oxidation and reduction: swelling and actuation

To fully oxidize the hydrogel rectangles prepared via Method 1 (0.2 mm thick), they were submerged in a 15 mM solution of CAN for 120 min, followed by washing with water (3x) and equilibration in water for ten min. The swelling caused by light-gated oxidation was determined by submerging the hydrogels in 30 mM aqueous iron(III)perchlorate for 30 min and irradiating the gel with 365 nm light for 2 h. After irradiation, the gels were washed with water (3x) and left to equilibrate in water for 10 min. To reduce the hydrogels, they were submerged in a 15 mM aqueous solution of ascorbic acid, followed by washing with water (3x) and equilibration in water for 10 min. The change in swelling was determined by measuring the change in the characteristic length of the rectangles (sum of the diagonals) using a USB microscope camera and the DinoCapture 2.0 software.

For the actuation experiments with the isotropic hydrogel, pieces of gel cut into strips (0.3 mm thick x 1.5 mm wide x 8–16 mm long) were soaked in a 30 mM solution of iron(III)perchlorate for 30 min. Then, for the actuation experiment, the hydrogel strip in the 30 mM iron(III) perchlorate solution was placed in a quartz cuvette positioned on its side. The cuvette was irradiated with 365 nm light (Thorlabs M365L3, 56 mW cm⁻²) from the bottom until the material stopped bending (~40 min).

Actuation of the hydrogel spring and curl was achieved by submerging the spring or curl in a 30 mM solution of CAN until there was no more motion observed. Then, the CAN solution was exchanged with water. Reduction using 15 mM aqueous ascorbic acid resulted in recovery of the initial curl/spring shape.

## Photopatterning

The hydrogel pieces (any dimensions, freestanding or on glass) cut into strips were soaked in a 30 mM solution of iron(III)perchlorate for 30 min. They were then removed from the solution and covered with a cover slip and mask (metal mask or laser printed on transparency) and irradiated with 365 nm light for up to 30 min (Thorlabs M365L3, 56 mW cm⁻²).

## Rheology

Rheological tests were performed on a TA Instruments Discovery HR-2 rheometer at 20 °C. An 8 mm crosshatched stainless steel Peltier plate was used. The gap was determined based on when there was contact between the top plate and the gel (80 μm for the neutral gel and 120 μm for the oxidized gel). A frequency sweep was done at 1% strain from 0.1 rad s⁻¹ to 100 rad s⁻¹, followed by an oscillation amplitude sweep at 5 rad s⁻¹ from 0.1% strain to 200% strain and the reverse (200% to 0.1%). The reported G' and G" are averages of three samples.

## Electrochemistry

All electrochemical experiments were carried out using a three-electrode setup using a PalmSens4 potentiostat. Pt wire was used as a counter electrode throughout. As a working electrode, a GC disk (3 mm diameter), a Pt disk (1.6 mm diameter), a Pt microdisk (10 μm diameter) or ITO (functionalized with BTX-gel) were used. The disk electrodes were polished using a 0.05 μm alumina slurry prior to each experiment. A non-aqueous Ag/AgNO₃ reference electrode (10 mM AgNO₃ in CH₃CN, 100 mM TBAPF₆) was employed for measurements in CH₂Cl₂ and all potentials were referenced against Fc as an external standard. In water, an Ag/AgCl (sat. KCl) reference electrode was used. All experiments were carried out under ambient conditions in HPLC-grade solvents and in the presence of oxygen. TBAPF₆ and NaClO₄ were used as supporting electrolytes in CH₂Cl₂ and water, respectively (100 mM in both cases). Controlled-potential coulometry was carried out in a divided bulk electrolysis cell (ALS Japan) with a porous carbon working electrode.

## Profilometry

Samples prepared via Method 3 using a 100 μm spacer were used for the profilometry measurements. The line profile scans and 3D mapping of the surface topography were performed using a DektakXT Bruker profilometer with a stylus tip radius of 12.5 μm and 1 mg of stylus force.

## Computational Methods

In order to theoretically study the switching properties of **BTX-MA** and **BTX-gel**, the simpler precursor featuring only two methoxy substituents **1** was chosen. This model compound allows for a reduction in the flexibility of the molecule, and thus in the number of possible conformers.

Initial geometries for the neutral (E/Z)-syn-, (E/Z)-anti- and (E/Z)-twisted isomers of **1**, as well as (E/Z)-twisted isomers of **1⁺** were generated using CREST[65–67] (Conformer–Rotamer Ensemble Sampling Tool) at the GFN2-xTB[68] level of theory. In order to compare the relative stability of each of these isomers, their geometry was further optimized with the composite method r²SCAN-3c[69], using the Orca 6.0.1 package[70]. The neutral forms of the (E/Z)-syn- and (E/Z)-anti-isomers were employed as initial guesses for the optimization of their radical cationic forms. All calculations were performed in the gas phase, and the counterions were not included. For all isomers, the obtained minima had no imaginary frequency. This methodology has been previously shown to accurately describe the relative stability of the various isomers of similar compounds[57].

Valence orbital energies were obtained by re-optimizing the structures of the (E/Z)-syn- and (E/Z)-anti- isomers of **1** at the ωB97X-D4[61,62]/def2-TZVPD[63] level of theory, which provides higher numerical accuracy. All calculations were performed in the gas phase, and the counterions were not included. For all isomers, the obtained minima had no imaginary frequency.

## Data availability

The data that support the findings of this study are available in the supplementary material of this article and are available from the

corresponding authors upon request. Coordinate files for calculated structures are provided. Source data are provided in this paper.

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

## Acknowledgements

R.R.W. acknowledges financial support from the EU (H2020 ITN grant "Art-MoMa" no. 860434). R.H. gratefully acknowledges financial support through a Marie Skłodowska-Curie Postdoctoral Fellowship (project number 101063933), a Liebig Fellowship (FCI) and a Return Fellowship by the state of North Rhine-Westphalia. Y.G. acknowledges financial support from a Marie Skłodowska-Curie Postdoctoral Fellowship (project number 101060079) and the Région Bretagne (SAD 2023_CNRS_PITCH). This work was supported by the Netherlands Organization for Scientific Research (B.L.F.), the Royal Netherlands Academy of Arts and Sciences (B.L.F.), the Dutch Ministry of Education, Culture, and Science (Gravitation Program 024.001.035 to B.L.F.) and the European Research Council (Advanced Investigator Grant 694345 to B.L.F.). This project was provided with computing HPC and storage resources by GENCI at IDRIS, thanks to the grant 2025-AD010816165 on the supercomputer Jean Zay's CSL partition (Y.G.). R.R.W. acknowledges V.B. Verduijn for help with operating the rheometer.

## Author contributions

R.H. and B.L.F. guided and supervised the research. R.R.W., R.H., A.R., and B.L.F. conceptualized the general idea of responsive BTX-gels and designed all experiments. R.H. conceptualized the light-gated redox switching approach. Y.G. carried out all theoretical calculations. D.G.R. conducted profilometry measurements under the guidance of M.A.L. R.H. carried out all electrochemical experiments. R.R.W. conducted all other experiments, including syntheses, material characterization and switching studies and prepared all figures. R.R.W. and R.H. prepared the manuscript, which was edited by A.R. and B.L.F. All authors have given approval to the final version of the manuscript.

## Competing interests

The authors declare no competing interests.
