## [Peer Review file · Nature Communications]

Light-gated Redox Switching and Actuation in Polymer Hydrogels

Corresponding Author: Professor Ben Feringa

Version 0:

Reviewer comments:

Reviewer #1

(Remarks to the Author)

This work reports the design, synthesis and preparation of a novel overcrowded alkene bithioxanthyllidene (BTX)-containing hydrogels that can undergo light-gated redox switching. The photophysical properties of the hydrogel have been studied with UV-vis absorption, and emission spectroscopies, while their electrochemical properties have also been studied with cyclic voltammetry. The physical properties of the swelling and actuation motion of the BTX-containing hydrogels have also been demonstrated. While the work can be regarded as an interesting example of a novel class of stimuli-responsive material with elements of novel concepts, the following points should be addressed before it is considered for publication.

1. Regarding the characterization of the compounds, elemental analysis data should be included.
2. Concerning the ^{13}C NMR experiment of compound 2, could the authors try to acquire the ^{13}C NMR spectrum using other deuterated solvents? Alternatively, given that compound 2 in CDCl_3 gives a well-resolved ^1H NMR spectrum with satisfactory signal-to-noise ratio, the authors should record the ^{13}C NMR spectrum in the same solvent with longer acquisition time.
3. The chemical reduction of the dicationic state was suggested to give the anti-folded state of the molecule (Figure S10). On the other hand, the electrochemical reduction showed the preference for the generation of the syn-folded state (Figure 1A). The authors should comment on this.
4. Regarding the electrochemical study, additional experiments should be done to confirm the two-electron oxidation process using controlled potential coulometry. The cyclic voltammograms clearly indicated irreversible redox waves. But in the description, there are no side products. This is not convincing. Is there an EC mechanism? Can one increase the scan/sweep rate using microelectrodes to go faster than the side reaction to get a more quasi-reversible redox couple? Can one exclude an EC mechanism occurring?
5. In Figure 3D–F, the arrows indicating the direction of the scan are not displayed properly.

Reviewer #2

(Remarks to the Author)

This work presents a creative approach for combination of both photo- and redox-switching in a single material. The molecular design and photoredox switching are both intriguing. I believe that this light-gated actuation of polymeric hydrogels will attract widespread attention and contribute significantly to the development of deformable materials. However, before publication, the authors need to address the following issues:

1. There are several scientific inaccuracies. For example, in Line 113: "Alternatively, in Method 3, free-standing polymer gels were prepared by polymerization induced by 365 nm light." However, according to Figure 2, the polymeric gels in Method 3 were initiated by 455 nm light. This discrepancy needs to be corrected.
2. Do the authors use the same images in Figure 2B and Figure S23? If so, this should be avoided.
3. The logical flow of the main text is somewhat confusing. The authors repeatedly disrupt the order in which the figures and diagrams are introduced, which may confuse the reader. A clearer sequence in the narrative would improve readability.
4. The coordination interactions between BTX-gel and ions should be clarified, especially regarding the changes in the network structure before and after redox switching. It would be helpful to show the coordination behavior with other ions to provide a more comprehensive understanding of the system.
5. The mechanism behind the light-gated redox condition, particularly the decrease in Eox (Figure 1A), could be discussed in greater detail to enhance the clarity of the process.

6. The actuating dynamics shown in Figure S26 should include error bars to quantify the variability in the data.
7. The dynamic interplay between photo-redox switching and swelling-induced bending (Figure 5) should be further discussed. It should be clarified whether UV light is applied throughout the deformation process, or only at the beginning, to better understand the timing and impact of light exposure on the system.
8. Regarding the patterning resolution shown in Figure 6, further explanation is needed on how the growth height is regulated.

Reviewer #3

(Remarks to the Author)

The authors describe a nice strategy to enable light and chemically triggered redox hydrogels. There are novel elements in the paper but the following issues need to be addressed.

a) Redox hydrogels are not new and the researchers need to better describe state of the art and also what the novelty and competitive advantage of their innovation is? Could the authors for example quantitatively compare features of the gel such as speed of actuation, concentration dependence, reusability / reversibility etc compared to other redox hydrogels.

b) The authors use "large" or "significant" swelling but for someone working in this area the swelling is not that large. If I understood this correctly swelling from 1x to 1.49 x (49%) is not that significant. For example polyelectrolyte DNA polymerization gels can swell about 300% and NIPAM / PAA etc can swell even more. The authors should compare their swelling ratio to other gels and especially polyelectrolyte and redox gels to qualify their statements. Also what are the limits to this swelling ratio and could it be increased further.

c) The figures look like one-of figures; the authors should include statistics and quantitative data. For example, how many samples were tried for Figure 5b or 6c, what was the variation in curvature; what about the time scale and standard deviation.

d) Could the authors discuss the swelling mechanism. Does it depend on the solvent characteristics (pH, ionic strength) and are there limitations of the approach. For example, are these chemicals toxic? What about the stability in storage and reusability? This is important from a practical perspective especially since the authors motivate use in soft-robotics (which arguably interfaces with humans)

Version 1:

Reviewer comments:

Reviewer #1

(Remarks to the Author)

The authors have addressed most of the concerns and the manuscript is ready for publication.

Reviewer #2

(Remarks to the Author)

[Editor's Note: This reviewer provided comments as note to the editor and does not have any further technical concerns]

Reviewer #3

(Remarks to the Author)

The authors have satisfactorily addressed issues and the paper can be accepted. While the chemicals are toxic for use in practical applications (as pointed out by the authors) in soft robotics, there is some value in the intellectual contribution to the mechanism of operation.

Detailed Response to Reviewer Comments

“Light-gated Redox Switching and Actuation in Polymer Hydrogels”

Roza R. Weber, Robert Hein*, Alexander Ryabchun, Yohan Gisbert, David Garcia Romero, Maria Antonietta Loi, and Ben L. Feringa*

Manuscript ID: NCOMMS-25-10039

We thank all reviewers for their positive response, scholarly evaluation and valuable comments regarding our manuscript, which we have improved according to their suggestions, as detailed in the following. All significant changes in the main text as well as the SI are highlighted in yellow.

We have also updated the list of authors and affiliations. Specifically, we have added Dr. Yohan Gisbert as a new author, as he made significant contributions in the form of detailed theoretical calculations as detailed below.

Reviewer #1:

This work reports the design, synthesis and preparation of a novel overcrowded alkene bithioxanthyllidene (BTX)-containing hydrogels that can undergo light-gated redox switching. The photophysical properties of the hydrogel have been studied with UV-vis absorption, and emission spectroscopies, while their electrochemical properties have also been studied with cyclic voltammetry. The physical properties of the swelling and actuation motion of the BTX-containing hydrogels have also been demonstrated. While the work can be regarded as an interesting example of a novel class of stimuli-responsive material with elements of novel concepts, the following points should be addressed before it is considered for publication.

1. Regarding the characterization of the compounds, elemental analysis data should be included.

Reply: We have recorded EA data for the monomer title compound, which confirms its structure/purity. This data has been included in the synthesis section (SI).

2. Concerning the ^{13}C NMR experiment of compound 2, could the authors try to acquire the ^{13}C NMR spectrum using other deuterated solvents? Alternatively, given that compound 2 in CDCl_3 gives a well-resolved ^1H NMR spectrum with satisfactory signal-to-noise ratio, the authors should record the ^{13}C NMR spectrum in the same solvent with longer acquisition time.

Reply: Thank you for this suggestion. We tried to solubilize this intermediate compound in several deuterated solvents at sufficient concentration, such as DCM, methanol, acetonitrile, DMSO, CDCl_3 , THF, toluene and a few mixtures of solvents such as DCM/MeOH. However, as we did not manage to record a satisfactory ^{13}C spectrum with any of the solvents that we tried, we ended up not including a ^{13}C spectrum in the SI. However, based on all other analyses (^1H NMR and MS), as well as the clear and full characterization of the final compound (see also comment above), we are confident that we have indeed obtained compound 2.

3. The chemical reduction of the dicationic state was suggested to give the anti-folded state of the molecule (Figure S10). On the other hand, the electrochemical reduction showed the preference for the generation of the syn-folded state (Figure 1A). The authors should comment on this.

Reply: Thank you for pointing this out. As discussed in more detail in the new, extensive section on the mechanism of the (redox) switching (SI Section 4.2, pages S13-21), reduction indeed quantitatively forms the *syn*-folded state first, which however quickly relaxes to the anti-folded state. We added the following text in the figure caption to make this clearer:

“Initially the syn-folded state is quantitatively formed which then thermally relaxes to the anti-folded state; in the depicted wavelength region neither of these species display any absorbance and due to the time scale of the experiment $BTX\text{-}gel_{anti}$ will be the predominant species.”

4. Regarding the electrochemical study, additional experiments should be done to confirm the two-electron oxidation process using controlled potential coulometry. The cyclic voltammograms clearly indicated irreversible redox waves. But in the description, there are no side products. This is not convincing. Is there an EC mechanism? Can one increase the scan/sweep rate using microelectrodes to go faster than the side reaction to get a more quasi-reversible redox couple? Can one exclude an EC mechanism occurring?

Reply: We realize that we did not discuss the specific redox switching mechanism in sufficient detail and have added a new Section in the SI (SI Section 4.2, pages S13-21), in which we present detailed theoretical and experimental analyses on various relevant switching aspects. The full new discussions/data are also reproduced in the appendix below. To briefly sum this up, from this work, and our previous published papers (e.g. 10.1021/jacs.4c08284), we have a clear understanding of all redox steps. Indeed, in the oxidative direction there is an ECE mechanism, whereby the chemical step is an important geometric rearrangement that gives rise to significant potential inversion and virtually simultaneous two-electron redox. We have proven this by additional data using chronoamperometry at micro-disk electrodes, controlled-potential bulk electrolysis as well as detailed theoretical calculations.

The reviewer’s suggestion of using high scan rates to observe other steps or render the process more reversible is very good, however even at the highest experimentally attainable scan rate of 250 V/s we did not observe quasi-reversible behavior, which indicates that the kinetics of the C step are very fast.

It is important to note that, in spite of the chemical irreversibility of the EC step, the overall redox process displays a very high degree of practical reversibility, i.e. repeat interconversion of the neutral and dicationic states without significant decomposition/formation of other side-products.

We have also added the following explanatory new text in the main text of the manuscript (pages 7-8):

“Transfer of two electrons upon oxidation was experimentally supported by Shoup-Shabo analysis⁵⁶ of the chronoamperometric response of a micro-disk electrode ($r = 5 \mu\text{m}$), as detailed in the SI, Section S4.2, Figures S27-28. Additional evidence for simultaneous two-electron oxidation was also obtained by bulk controlled-potential electrolysis, which confirmed that both

the oxidation and the reduction waves comprise two electrons (Figure S29). As discussed in more detail in the SI Section 4.2, and in the literature^{57,58}, the two-electron oxidation from the anti-folded state to the orthogonal dicationic state proceeds via an E_1CE_2 mechanism with strong potential inversion, i.e. $E_2 < E_1$, Scheme S1), with a very fast chemical rearrangement in the intermediate radical cation state. This was supported by calculations of the optimized geometries of all relevant isomers and their relative energies (Figures S15-26). These theoretical investigations also confirmed that both the E and Z isomers of the BTX-switch display very similar geometries and energies and are thus expected to switch in the same manner.

As a result of the significant geometric rearrangements that are associated with electron transfer, the redox process displays a very high degree of hysteresis; two-electron reduction of the dication back to the neutral state occurs at much more cathodic potentials (-0.05 V vs. Fc/Fc^+ via an EEC pathway, which first quantitatively forms the syn-folded state, Scheme S1), ...”

5. In Figure 3D–F, the arrows indicating the direction of the scan are not displayed properly.

Reply: Thank you, we have updated the figure.

Reviewer #2:

This work presents a creative approach for combination of both photo- and redox-switching in a single material. The molecular design and photoredox switching are both intriguing. I believe that this light-gated actuation of polymeric hydrogels will attract widespread attention and contribute significantly to the development of deformable materials. However, before publication, the authors need to address the following issues:

Reply: We thank the reviewer for their positive comments on our work and valuable suggestions.

1. There are several scientific inaccuracies. For example, in Line 113: “Alternatively, in Method 3, free-standing polymer gels were prepared by polymerization induced by 365 nm light.” However, according to Figure 2, the polymeric gels in Method 3 were initiated by 455 nm light. This discrepancy needs to be corrected.

Reply: Thank you for pointing out this typing error. We in fact meant to write “Alternatively, in Method 2, free-standing polymer gels were prepared by polymerization induced by 365 nm light.” We have fixed this mistake.

2. Do the authors use the same images in Figure 2B and Figure S23? If so, this should be avoided.

Reply: Indeed, parts of this figure are reproduced in the SI, however at a much larger amplification and containing additional details (i.e. size scales), that allow a better visualization of the material. We believe this is valuable to the reader and have accordingly not made any changes.

3. The logical flow of the main text is somewhat confusing. The authors repeatedly disrupt the order in which the figures and diagrams are introduced, which may confuse the reader. A clearer sequence in the narrative would improve readability.

Reply: We have updated the order of the panels in Figure 1 to correspond to the order in which they are discussed in the main text.

4. The coordination interactions between BTX-gel and ions should be clarified, especially regarding the changes in the network structure before and after redox switching. It would be helpful to show the coordination behavior with other ions to provide a more comprehensive understanding of the system.

Reply: Thank you for this suggestion. We have conducted various additional experiments in triplicate to gain more insights into the effect of the counter anions on switching and swelling (Figures S40-43). Specifically, we studied the effect of the nature of the counter anion (NO_3^- , PF_6^- or ClO_4^-), their concentration (0, 1, 10 or 100 mM) as well as the pH (1,3,4 and 7, all in 100 mM NaNO_3) both prior to and after chemical redox switching with CAN. These conditions were chosen to represent a large range of conditions without compromising the system (e.g. the counter anion must be compatible with the oxidative environment, which e.g. precludes halides). Gratifyingly, under all tested conditions, chemical oxidation proceeded smoothly and led to significant swelling (~30%), which is very similar to the swelling under the “standard”

conditions (i.e. in pure water). In general, there are only relatively small differences in swelling degree upon changing conditions, which however follow predictable trends which are related to ion concentration (slightly less swelling at higher ionic strength). We added discussions about this in the SI (page S29-32, see also Appendix). We also added the following in the main text (page 10):

“Furthermore, successful oxidation was achieved in a wide variety of aqueous environments containing different counter anions/salts (NaPF₆, NaNO₃ or NaClO₄) at different concentrations (0 – 100 mM) and different pH (1 – 7), see Figures S40-42 and associated discussions. In general, we observed no strong influence of differing conditions on the swelling magnitude, which was around 20 – 30% in all cases. The only consistent, but weak trend we identified is that a higher ionic strength induces somewhat diminished responses, which is in good agreement with Donnan theory.⁶⁰ These observations suggest that swelling is primarily driven by the generation of the dicationic charge state, that specific ion pairing effects (i.e. the nature of the counter anion) are not overly important and that the BTX gel actuators can be used under various aqueous conditions and in different application scenarios. Moreover, as shown in Figure S33, adjusting the amount of BTX-MA or crosslinker has a comparably much larger effect on the observed magnitude of redox-induced swelling, but can also alter the inherent mechanical properties of the gel.”

5. The mechanism behind the light-gated redox condition, particularly the decrease in E_{ox} (Figure 1A), could be discussed in greater detail to enhance the clarity of the process.

Reply: We thank the reviewer for pointing this out. This actually turned out to be a much more complicated, and interesting, open question than we anticipated. Initially, we thought that this is simply related to the ease via which the first electron is removed from the neutral *anti*- or *syn*-folded states, i.e. an increase in HOMO energy. To verify this, we conducted in-depth DFT calculations of the HOMO (and LUMO) energies via multiple different methods for both the E and Z isomers of the switching core. In all cases we observed that the difference in HOMO energies is insufficient to explain the altered oxidation potential, which indicates that other mechanisms are responsible for this. Specifically, this could be related to the kinetics of the “C” step or a complete change in oxidation mechanism (e.g. CEE), however this requires further in-depth electrochemical and theoretical studies, which are beyond the scope of this work, but will be investigated in our laboratories.

All of this new data and discussions are included in the SI (Section S4.3, pages S21-25, see also Appendix below).

We also added the following new paragraph in the main text (page 10):

“The exact origin of the change of oxidation potential between the anti- and syn-folded states has not yet been elucidated. We initially hypothesized that the facilitated oxidation for the syn-folded state is related to an increase in its HOMO energy and thus ionization potential. However, DFT calculations at the ω B97X-D4^{61,62}/def2-TZVPD⁶³ level of theory suggest that the difference between HOMO energies is too small to explain the observed magnitude of potential shift (see Figures S30-31, Tables S2-3 and associated discussions). This indicates that other mechanisms, e.g. related to kinetics of conformational rearrangements, are relevant.”

6. The actuating dynamics shown in Figure S26 should include error bars to quantify the variability in the data.

Reply: We have conducted additional repeat measurements on three independent gel strips of the same dimensions, which all show very similar actuation dynamics ($t_{1/2} = 1.1$ min), as can be seen by the low error bars (see Figure S48). In the original Figure S26

(now Figure S47), it can also be seen that even for slightly differently sized strips the dynamics are virtually identical.

7. The dynamic interplay between photo-redox switching and swelling-induced bending (Figure 5) should be further discussed. It should be clarified whether UV light is applied throughout the deformation process, or only at the beginning, to better understand the timing and impact of light exposure on the system.

Reply: Thank you for pointing this out, we made this clearer in various places in the text how the irradiation was carried out. In the majority of cases, we carried out continuous irradiation until the motion/swelling stopped, i.e. the full extent of switching (e.g. Figures 5 and 6, Figures S46-48). In the captions of the figures we also stated more clearly that the light exposure was continuous until the bending stopped.

However, it is of course possible to achieve intermediate conversion/actuation by shorter irradiation. To study this in more detail, we carried out additional experiments on the light-gated redox bending of a gel strip, see the new Figure S49. These experiments further highlight that the extent of actuation can be controlled by irradiation duration, but that the swelling response lags somewhat behind the photoswitching. We have also added the following discussion in the main text (page 12):

“Furthermore, the extent of light-gated redox switching, and the resulting swelling/motion can be controlled by adjusting the irradiation duration. For example, as shown in Figure S49, three different levels of bending were achieved by interrupted irradiation with short bursts of light. Notably, there is a lag period between the short irradiation and the full extent of the resulting actuation, which can be attributed to the comparably slower diffusion-controlled redox/swelling process. Nevertheless, these results indicate that any desired degree of light-gated redox switching can be achieved by controlling the overall photon flux.”

8. Regarding the patterning resolution shown in Figure 6, further explanation is needed on how the growth height is regulated.

Reply: For the patterning experiment, we irradiate for a relatively long period of time (30 min) because we found that this ensures that the maximum possible amount of the dication is generated in the material, so that we achieve the largest swelling effect. The height that is shown in Figure 6, is the height that resulted from that treatment. As discussed above, intermediate states are accessible. We have added the following text in the caption of Figure 6 to clarify this:

“Irradiation was carried out for ~30 min to ensure full conversion and the largest extent of swelling. However, in principle, any intermediate swelling state can be accessed by shorter irradiation duration.”

Reviewer #3:

The authors describe a nice strategy to enable light and chemically triggered redox hydrogels. There are novel elements in the paper but the following issues need to be addressed.

Reply: We thank the reviewer appreciating our strategy and for their supportive evaluation and suggestions provided.

a) Redox hydrogels are not new and the researchers need to better describe state of the art and also what the novelty and competitive advantage of their innovation is? Could the authors for example quantitatively compare features of the gel such as speed of actuation, concentration dependence, reusability / reversibility etc compared to other redox hydrogels.

Reply: Thank you for this suggestion, we agree that some additional comparisons are useful. We have added the following text on page 9:

“To put this value into context, previous work on redox-active hydrogels, such as the materials developed by Harada and coworkers in 2013, which are based on reversible host-guest interactions to swell and shrink, showed an oxidation-induced swelling of only 11% in length.⁴⁶ Other representative redox-responsive gel actuators were reported in 2023 by Ikeda, which were based on foldamers. These organogels, which were not operated in water but in acetonitrile, swelled up to 25% when oxidized.⁴⁵ A different approach to generate responsive hydrogels, which is also based on light and redox stimuli, makes use of photoredox catalysts. This strategy was notably developed by Barnes and coworkers, wherein irradiation of a separate photocatalysts induced local reduction and shrinking of viologen-foldamer gels.^{47,49,59} However, this had to be performed in an oxygen-free environment and induced a slightly smaller decrease in volume of 88%, which corresponds to a 23% decrease in length.⁴⁹ Exposure to oxygen then induced oxidation and led to the recovery of the initial gel size. Of additional note is that in all of the examples discussed above, only 2-3 full redox switching cycles were shown.”

b) The authors use "large" or "significant" swelling but for someone working in this area the swelling is not that large. If I understood this correctly swelling from 1x to 1.49 x (49%) is not that significant. For example polyelectrolyte DNA polymerization gels can swell about 300% and NIPAM / PAA etc can swell even more. The authors should compare their swelling ratio to other gels and especially polyelectrolyte and redox gels to qualify their statements. Also what are the limits to this swelling ratio and could it be increased further.

Reply: Thank you for pointing this out; indeed the swelling by simply changing the solvent (here from EtOH to water) is, in comparison to other systems, not necessarily particularly large. We have removed the word “significant” to avoid confusion. As this work is not focused on the swelling of the neutral gel, but swelling induced by oxidation, we have not carried out further in-depth studies of the neutral swelling, apart from the influence of salt/pH, see SI Section 6, Figures S40-42, which revealed no significant effects of the ionic strength. The swelling upon oxidation is somewhat more tunable by ionic strength, however the most straight-forward way to tune this is via adjusting the amount of the BTX and/or crosslinker in the gel (see Figure S33), however this also alters the mechanical properties of the materials. We added the following sentence in the main text (page 10) to draw attention to this:

*“Moreover, as shown in Figure S33, adjusting the amount of **BTX-MA** or crosslinker has a comparably much larger effect on the observed magnitude of redox-induced swelling, but can also alter the inherent mechanical properties of the gel.”*

c) The figures look like one-of figures; the authors should include statistics and quantitative data. For example, how many samples were tried for Figure 5b or 6c, what was the variation in curvature; what about the time scale and standard deviation.

Reply: We conducted numerous repeats of the switching/swelling experiments (light, redox and light-gated redox) and observed a very high degree of reproducibility in all cases. We have made this clearer in all figures where repeats were carried out (majority of figures in the SI). In some cases the error bars are actually so small that they are barely visible (e.g. Figure S10). We also carried out additional bending experiments of the gel strips shown in Figure 5b, which are very reproducible, even for strips of slightly different sizes (Figures S47-48). All swelling experiments at different salt concentrations (S40-42) were also carried out in triplicates and show low errors.

d) Could the authors discuss the swelling mechanism. Does it depend on the solvent characteristics (pH, ionic strength) and are there limitations of the approach. For example, are these chemicals toxic? What about the stability in storage and reusability? This is important from a practical perspective especially since the authors motivate use in soft-robotics (which arguably interfaces with humans)

Reply: We expect swelling to be driven by repulsive electrostatic interactions between the switches as well as the Donnan effect. The former is probably strongly solvent dependent, which was not studied herein, but may provide an additional means of tuning performance; we have added information on this in the main text (page 9):

“This is consistent with electrostatic repulsion between the switches as well as the larger hydrophilicity of the oxidized, highly charged gel state which results in significant uptake of more water into the gel (Donnan effect), both of which are potentially also tunable by changing solvents/ion concentrations (vide infra).”

As detailed in response to reviewer 2, point 4, we have also added additional data and discussions of pH and ionic strength on swelling.

Regarding the toxicity: The chemical oxidants are both rather strong oxidizers and irritants/toxic, so interfacing with humans is currently not advisable. However, we believe that the principle of light-gated redox switching can be extended in the future to be interfaced with electrodes or use more benign chemicals, e.g. ambient oxygen. We have made this clearer in the conclusions section:

“...can be individually carried out and further tuned by chemical modification of the (BTX) switch⁵⁷, for example such that weaker, less toxic oxidants – ideally ambient oxygen – can be employed.”

Regarding the stability/reusability: As shown in Figure S10C, the reversibility of the redox switching is already good, but can clearly be improved in the future. The stability of the gels overtime is very good in both charge states (see also new Figure S43, we have added the following on page 10:

*“As a result, the oxidized **BTX-gel**²⁺ state can, in principle, be maintained indefinitely; by the naked eye we observed only minor changes in colour/morphology of the **BTX-gel**²⁺ after one month in aqueous solution (Figure S43).”*

Appendix

4.2 Mechanism of Redox Switching

Starting from the neutral, *anti*-folded state, one electron oxidation occurs at a high potential ($E_{Af,1}$) and transiently generates a radical cation in the same conformational state ($Af^{\bullet+}$, Scheme S1). This species very quickly chemically rearranges to its most preferred conformational state ($Tw^{\bullet+}$, via k_1). This twisted radical cation now possesses a *lower* oxidation potential for removal of the second electron ($E_{Tw,2}$, with $E_{Tw,2} \ll E_{Af,1}$, i.e. potential inversion), such that the dicationic species in the twisted/orthogonal state (Ort^{2+}) is immediately generated (any redox potential sufficient to induce the first $E_{Af,1}$ process will automatically force the second electron transfer, such that both electrons are in effect transferred simultaneously and the intermediate radical cation $Tw^{\bullet+}$ is not formed in any significant amounts, giving rise to a simultaneous two-electron oxidation at ~ 0.8 V, see Figure 3 main text). In the forward direction (*anti*-folded \rightarrow dication) the redox switching thus occurs via an ECE mechanism with a chemically irreversible first step, “ratcheted” by the favourable and fast geometric rearrangement of $Af^{\bullet+}$ to $Tw^{\bullet+}$). This was further supported by calculations of the relative conformer energies at the r^2 SCAN-3c level of theory, which confirm that for the radical cation charge state, the $Af^{\bullet+}$ conformer is ~ 30 kJ/mol higher in energy than $Tw^{\bullet+}$ (see Table S1), such that the first one-electron transfer is (rapidly) followed by a chemical rearrangement step.

Af – anti-folded

Syn – syn-folded

Tw – twisted

Ort – orthogonal

Supplementary Scheme 1: Detailed square scheme depicting the relevant redox (green arrows) and structural (blue arrows) interconversions between all relevant redox states. The thermodynamically most stable and most

relevant species for each oxidation state are highlighted with a box and for the neutral and radical cationic charge states the relative energies of all conformers are given in parentheses (energy differences are given relative to the lowest lying conformer of each redox state, at 25 °C (vide infra). For simplicity, most pathways are depicted as only proceeding in one direction; however, in principle, all redox and structural processes are (reversible) equilibria; most depicted ET processes are rendered chemically irreversible through fast, subsequent rearrangements.

In principle, the second electron transfer between $Tw^{+\bullet}$ and Ort^{2+} is a fully reversible step as it occurs without any significant geometric rearrangements, or at least with very low barriers (Note that while for the BTX^{2+} dication, a perpendicular arrangement of the rotors has been confirmed crystallographically,^[1] other more “twisted” conformations are energetically easily accessible by rotation around the central single-bond; in fact for a related derivative we recently crystallographically observed a 64° angle between the mean planes of the two rotors in the dicationic “orthogonal” state).^[2]

In spite of this reversible interconversion, the $Tw^{+\bullet}$ state can still not be accessed by reduction of the dication (at least not for the homomeric BTX derivatives^[2]). This is because at potentials at which the reduction can be achieved (~ -0.05 V, see Figure 3 main text), the reduction of the second thioxanthylum rotor also occurs as there is very little electronic communication between the two halves in the single-bonded orthogonal state (i.e. $E_{Tw,1} \sim E_{Tw,2}$, potential compression). As a result, during reduction simultaneous transfer of two electrons occurs easily, which transiently generates the neutral twisted state (Tw), which is an unfavored conformation for this charge state and thus very quickly rearranges to the *syn*-folded state (k_2). This was again confirmed by DFT calculations; for the neutral switch, Tw (~ 60 kJ/mol) is much higher in energy than the *syn*- (~ 44 kJ/mol) or *anti*-folded states ($\equiv 0$ kJ/mol). The exclusive formation of *Syn* by reduction of Ort^{2+} (via the very unstable Tw) was confirmed previously.^[1, 3] As discussed in the main text, *Syn* can then thermally relax to *Anti* (k_4), which is however much slower than the other chemical rearrangements ($t_{1/2} \sim 150$ s). As a result *Syn* can be quantitatively populated by irradiation and also oxidized, most likely via an ECE mechanism, this time via the $E_{Syn,1}$ and k_3 pathway and the transient generation of the $Syn^{+\bullet}$ state, whose relative energy is with 74 kJ/mol even higher than that of $Af^{+\bullet}$ (~ 30 kJ/mol). It is important to note that, despite the chemical irreversibility of the EC and/or EEC steps, the overall redox process displays a very high degree of practical reversibility, i.e. repeat interconversion of the neutral and dicationic states without significant decomposition/formation of other side-products.

The calculated structures and energies of all the relevant conformational and charge states discussed above are shown in Table S1 and Figures S15-26. For simplicity these calculations, as well as the following electrochemical studies were carried out using the 2,2'-dimethoxy-BTX precursor **1** as a simpler analogue of the BTX-MA monomer. Substitution at one of the methoxy groups to give the monomer is not expected to affect the switching properties of the core, even more significant substitutions do not significantly alter the general (redox) switching properties of the system.^[3-4]

As shown in Table S1, the relative energies of the 2,2'-dimethoxy-BTX precursor **1** were calculated for both the *E* and *Z* isomers of all conformers and both the neutral and monocationic charge state. In all cases the energy difference between the *E* and *Z* isomers is negligible, confirming that their (redox) switching behavior is the same, regardless which isomer is used (note we herein always use a mixture of both isomers).

Supplementary Table 1: Calculated Gibbs free energy differences (kJ/mol) at 25 °C of the different isomers of **1**, relative to the energy of the lowest isomer within each redox state. Calculations were performed at the r²SCAN-3c level of theory in the gas phase.

	Neutral	Radical Cation
E-anti -folded	0	29.5

E-syn-folded	43.5	74.1
E-twisted	59.0	0
Z-anti-folded	0.6	29.6
Z-syn-folded	44.5	74.6
Z-twisted	60.0	0.1

Supplementary Figure 1: Optimized geometry of the *E-anti* isomer of **1** (neutral state).

Supplementary Figure 2: Optimized geometry of the *E-syn* isomer of **1** (neutral state).

Supplementary Figure 3: Optimized geometry of the *E-twisted* isomer of **1** (neutral state).

Supplementary Figure 4: Optimized geometry of the *Z-anti* isomer of **1** (neutral state).

Supplementary Figure 5: Optimized geometry of the *Z-syn* isomer of **1** (neutral state).

Supplementary Figure 6: Optimized geometry of the twisted *Z* isomer of **1** (neutral state).

Supplementary Figure 7: Optimized geometry of the *E-anti* isomer of **1**^{+•} (radical cationic state).

Supplementary Figure 8: Optimized geometry of the *E-syn* isomer of 1^{2+} (radical cationic state).

Supplementary Figure 9: Optimized geometry of the *E-twisted* isomer of 1^{2+} (radical cationic state).

Supplementary Figure 10: Optimized geometry of the *Z-anti* isomer of 1^{2+} (radical cationic state).

Supplementary Figure 11: Optimized geometry of the *Z-syn* isomer of 1^{2+} (radical cationic state).

Supplementary Figure 12: Optimized geometry of the *Z*-twisted isomer of 1^{**} (radical cationic state).

Number of transferred electrons

To further confirm that the single oxidative wave observed in the CVs corresponds to transfer of two electrons, we carried out the following additional experiments.

The time-dependent current response at a microdisk electrode following a potential step is given by the following equation as derived by Shoup and Szabo:^[5]

$$I = 4nFDcrf(\tau)$$

Where:

$$f(\tau) = 0.7854 + 0.8862 * \tau^{-0.5} + 0.214 * \exp [0.7823\tau^{-0.5}]$$

and:

$$\tau = \frac{4Dt}{r^2}$$

With c = bulk concentration, n = number of electrons transferred, D = diffusion constant, r = radius of the microdisk electrode and F = Faraday constant.

As shown by Compton and co-workers, both D and n can be simultaneously obtained by deconvolution of the current-response obtained from a simple chrono-amperometric experiment by non-linear curve fitting according to the equations above (when r and c are known).^[6]

This analysis was carried out herein for **1** via chronoamperometry of a 1 mM solution of **1** in CH_2Cl_2 , 100 mM TBAPF_6 using a Pt microdisk electrode ($r = 5 \mu\text{m}$) and OriginPro 2018. Figure S27 shows the CV under these conditions, while Figure S28 shows the CA current response.

Supplementary Figure 13: CV of 1 mM **1** in CH_2Cl_2 , 100 mM TBAPF_6 at a Pt micro disk electrode ($r = 5 \mu\text{m}$) at $v = 100 \text{ mV/s}$.

Analysis of this current response according to the equations shown above affords a good fit to the experimental data with $D = 9.8 \times 10^{-10} \text{ m}^2/\text{s}$ and $n = 1.82$, confirming that the oxidation wave of the BTX derivatives corresponds to transfer of two electrons.

Supplementary Figure 14: Chronoamperometric current response ($E = 1.01 \text{ V}$) for the oxidation of 1 mM **1** in CH_2Cl_2 , 100 mM TBAPF_6 at a Pt micro disk electrode ($r = 5 \mu\text{m}$) including a fit according to the Shoup-Szabo equation.

This was further supported by controlled-potential bulk electrolysis of **1** in 1:1 (v/v) $\text{CH}_3\text{CN}/\text{CH}_2\text{Cl}_2$, 100 mM TBAPF_6 in a divided bulk electrolysis cell, which again confirmed that the oxidation wave is comprised of two electrons (Figure S29). Additionally, this confirmed that reduction also proceeds via two electrons. The mixed solvent system containing 50% CH_3CN was chosen to ensure good solubility of both the neutral and dicationic charge state of **1**.

Supplementary Figure 15: Controlled-potential chronocoulometry of 13.1 μmol **1** in 1:1 (v/v) $\text{CH}_3\text{CN}/\text{CH}_2\text{Cl}_2$, 100 mM TBAPF₆ in a divided bulk electrolysis cell (WE: porous carbon, CE: Pt wire coil, RE: Ag/AgNO₃) confirming that both oxidation and reduction proceed via two electrons.

4.3 Mechanism for Change in Redox Potential Upon Irradiation

As mentioned in the main text, we hypothesized that the lowered oxidation potential of the *syn*-folded BTX switch state is related to its first ionization potential, i.e. its HOMO energy, for the first one electron oxidation ($E_{\text{Syn},1}$ and $E_{\text{Af},1}$ in Scheme 1). According to Koopmans' theorem,^[7] within the Hartree-Fock approximation, the first ionization energy of a molecule is equal to the negative of the energy of the HOMO. While this theorem is not exact for DFT-calculated energies, numerous studies report accurate predictions of ionization potentials using this method at the DFT level.^[8-9]

Figures S30-31 and Table S2-3 show the valence orbitals energy diagram and highest occupied orbitals at the $\omega\text{B97X-D4}/\text{def2-TZVPD}$ level of theory for both folded isomers of *E*-**1** and *Z*-**1**, respectively. As shown in Figures S30 and S31, the HOMO energies for all isomers (*E* and *Z* as well as *syn*- and *anti*-folded) are very similar.

Specifically, at the $\omega\text{B97X-D4}/\text{def2-TZVPD}$ level of theory, the following difference in ionization potential (IP) are obtained between the *syn*- and *anti*- forms of the *E*- and *Z*- isomers:

$$\Delta\text{IP}_{\text{anti/syn}}(E\text{-}\mathbf{1}) = \text{IP}_{\text{anti}}(E\text{-}\mathbf{1}) - \text{IP}_{\text{syn}}(E\text{-}\mathbf{1}) = -\epsilon\text{HOMO}(E\text{-anti-}\mathbf{1}) - (-\epsilon\text{HOMO}(E\text{-syn-}\mathbf{1})) = 7.7969 - 7.7901 = \mathbf{0.0068\text{ eV}}$$

$$\Delta\text{IP}_{\text{anti/syn}}(Z\text{-}\mathbf{1}) = \text{IP}_{\text{anti}}(Z\text{-}\mathbf{1}) - \text{IP}_{\text{syn}}(Z\text{-}\mathbf{1}) = -\epsilon\text{HOMO}(Z\text{-anti-}\mathbf{1}) - (-\epsilon\text{HOMO}(Z\text{-syn-}\mathbf{1})) = 7.7915 - 7.7751 = \mathbf{0.0164\text{ eV}}$$

This suggests that the *syn*- form of both *E* and *Z* isomers are slightly easier to oxidize than the *anti*-form, with a difference of 7 and 16 mV respectively. However, these differences are too small to explain the experimentally observed difference in oxidation potential, whereby the *syn*-folded state displays at

least 100 mV lower E_{ox} (Figure S13). This suggests that other factors need to be considered to explain the full extent of the observed difference in oxidation potentials. Specifically, this could be related to the kinetics of the “C” step or a complete change in oxidation mechanism (e.g. CEE), however this requires further in-depth electrochemical and theoretical studies, which are beyond the scope of this work.

Lastly, we would also like to highlight that these calculations correctly predict an increase in HOMO-LUMO gap upon light-induced switching to the syn-folded state, as is indeed observed experimentally by a blue-shift in the absorbance spectrum (Figure 3B).

Supplementary Figure 16: Valence orbitals energy diagram (LUMO-4 to HOMO+4) for the $E\text{-syn}$ and $E\text{-anti}$ isomers of **1** at the $\omega\text{B97X-D4/def2-TZVPD}$ level of theory.

Supplementary Table 2: Plot of the highest occupied orbitals for the $E\text{-syn}$ and $E\text{-anti}$ isomers of **1** ($\omega\text{B97X-D4/def2-TZVPD}$).

	$E\text{-syn}$	$E\text{-anti}$
HOMO		
HOMO-1		

Supplementary Figure 17: Valence orbitals energy diagram (LUMO-4 to HOMO+4) for the *Z-syn* and *Z-anti* isomers of **1** at the ω B97X-D4/def2-TZVPD level of theory.

Supplementary Table 3: Plot of the highest occupied orbitals for the *Z-syn* and *Z-anti* isomers of **1** (ω B97X-D4/def2-TZVPD).

	Z-syn	Z-anti
HOMO		HOMO-1		HOMO-2		HOMO-3		

Swelling experiments in the presence of various salts and pH

In order to study the influence of various salt and pH conditions on the swelling behaviour of the neutral gel, as well as for the chemical oxidation with CAN, the following experiments were carried out:

The size of the BTX-gel was first measured in pure water. The medium was then exchanged with the salt solution and after 30 minutes of equilibration the size was measured again. Afterwards, CAN was added (15 mM total, added as a stock solution of same salt concentration/pH) and the oxidized gels were equilibrated in the oxidant for 2 hours. The oxidized gels were then washed with pure salt solution and their size remeasured.

As shown in Figure S40, the nature of the counter anion (NO_3^- , PF_6^- or ClO_4^-) did not significantly alter the swelling properties of the gel, neither in the neutral state, nor after oxidation. Only for PF_6^- a slightly smaller swelling magnitude was observed after oxidation, however this can in part be attributed to some degree of precipitation formed by reaction between the CAN oxidant and the counter anion. Nevertheless, gel oxidation was still observed.

As shown in Figures S41 and S42, increasing the ionic strength (either by increase of the electrolyte concentration (NaNO_3) or by lowering the pH using HNO_3 (at a constant background concentration of 100 mM NaNO_3) induced a small decrease in the degree of swelling upon oxidation. For the swelling of the neutral gels upon exchange of the medium from pure water to the electrolyte, no clear trends that correlate with overall ionic strength were observed, however the change in gel size was comparably small in all cases (max $5 \pm 2\%$) for 100 mM NaNO_3 at pH 7.

Supplementary Figure 18: Swelling percentage (% increase in the characteristic length) of **BTX-gel** after the exchange of the swelling medium from water to a 100 mM sodium salt solution (off-white bars) and after oxidation using CAN (15 mM) in the same salt solution (purple bars), relative to the size of the gel in water. Upon addition of CAN to the gel in 100 mM NaPF₆ some precipitate was observed, however the oxidation still took place. Error bars represent one standard deviation of independent triplicate experiments.

Supplementary Figure 19: Swelling percentage (% increase in the characteristic length) of **BTX-gel** after the exchange of the swelling medium from water to NaNO₃ solutions with different concentrations (off-white bars)

and after oxidation using CAN (15 mM) in the same salt solution (purple bars), relative to the size of the gel in water. Error bars represent one standard deviation of independent triplicate experiments.

Supplementary Figure 20: Swelling percentage (% increase in the characteristic length) of **BTX-gel** after the exchange of the swelling medium from water to a 100 mM NaNO₃ solution with different pH (off-white bars) and after oxidation using CAN (15 mM, at the same pH; purple bars), relative to the size of the gel in water. The pH was adjusted using HNO₃. Error bars represent one standard deviation of independent triplicate experiments.

Supplementary Figure 21: Pictures of the BTX-gel in 100 mM NaNO₃ (top left), after oxidation with 15 mM CAN (top right) and after one month in the salt solution (due to some solvent evaporation the exact concentration of NaNO₃ is not known).

- [1] W. R. Browne, M. M. Pollard, B. de Lange, A. Meetsma, B. L. Feringa, *J. Am. Chem. Soc.* **2006**, *128*, 12412-12413.
- [2] R. Hein, C. N. Stindt, B. L. Feringa, *J. Am. Chem. Soc.* **2024**, *146*, 26275-26285.
- [3] R. Hein, Y. Gisbert, B. L. Feringa, *J. Am. Chem. Soc.* **2025**, *147*, 13649-13657.
- [4] B. P. Corbet, M. B. S. Wonink, B. L. Feringa, *Chem. Commun.* **2021**, *57*, 7665-7668.
- [5] D. Shoup, A. Szabo, *J. Electroanal. Chem. Interf. Electrochem.* **1982**, *140*, 237-245.
- [6] C. A. Paddon, D. S. Silvester, F. L. Bhatti, T. J. Donohoe, R. G. Compton, *Electroanalysis: An International Journal Devoted to Fundamental and Practical Aspects of Electroanalysis* **2007**, *19*, 11-22.
- [7] T. Koopmans, *physica* **1934**, *1*, 104-113.
- [8] J. Conradie, A Frontier orbital energy approach to redox potentials, in *Journal of physics: conference series*, Vol. 633, IOP Publishing, **2015**, p. 012045.
- [9] D. D. Méndez-Hernández, P. Tarakeshwar, D. Gust, T. A. Moore, A. L. Moore, V. Mujica, *J. Mol. Model.* **2013**, *19*, 2845-2848.